# Data-Driven Offline Decision-Making via Invariant Representation Learning

**Han Qi**\*, **Yi Su**\*, **Aviral Kumar**\*, **Sergey Levine**
Department of Electrical Engineering and Computer Sciences, UC Berkeley
{han2019, aviralk}@berkeley.edu, yisumtv@google.com   (\*Equal Contribution)

## Abstract

The goal in offline data-driven decision-making is synthesize decisions that optimize a black-box utility function, using a previously-collected static dataset, with no active interaction. These problems appear in many forms: offline reinforcement learning (RL), where we must produce actions that optimize the long-term reward, bandits from logged data, where the goal is to determine the correct arm, and offline model-based optimization (MBO) problems, where we must find the optimal design provided access to only a static dataset. A key challenge in all these settings is distributional shift: when we optimize with respect to the input into a model trained from offline data, it is easy to produce an out-of-distribution (OOD) input that appears erroneously good. In contrast to prior approaches that utilize pessimism or conservatism to tackle this problem, in this paper, we formulate offline data-driven decision-making as *domain adaptation*, where the goal is to make accurate predictions for the value of optimized decisions ("target domain"), when training only on the dataset ("source domain"). This perspective leads to invariant objective models (IOM), our approach for addressing distributional shift by enforcing invariance between the learned representations of the training dataset and optimized decisions. In IOM, if the optimized decisions are too different from the training dataset, the representation will be forced to lose much of the information that distinguishes good designs from bad ones, making all choices seem mediocre. Critically, when the optimizer is aware of this representational tradeoff, it should choose not to stray too far from the training distribution, leading to a natural trade-off between distributional shift and learning performance.

## 1   Introduction

Many real-world applications of machine learning involve learning how to make better decisions from data. When we must make a sequence of decisions (e.g., control a robot), this can be formulated as offline reinforcement learning (RL) [27, 30], whereas in problems where we must synthesize only one decision (e.g., a molecule that can effectively catalyze a reaction), this can be formulated as a multi-armed bandit or a data-driven model-based optimization (also known as black-box optimization [2]). In all of these cases, an algorithm is provided with data from previous experiments consisting of $(\mathbf{x}, y)$ tuples, where $\mathbf{x}$ represents a decision (e.g., a bandit arm, a design such as a molecule or protein, or a sequence of actions in RL) and $y$ represents its utility (e.g., certain property of the molecule or the long-term reward of a trajectory). The goal is to generate the best possible decision, typically one that is better than the best one in the dataset. All of these problem domains are united by a common challenge: in order to improve, the decisions learned by the algorithm must differ from the distribution of the designs in the training data, inducing *distributional shift* [23, 24], which must be carefully handled.

Current methods for solving such data-driven decision-making problems try to learn some form of a proxy model $\hat{f}(\cdot)$ that maps a decision $\mathbf{x}$ to its objective value $y$ via supervised learning on the

36th Conference on Neural Information Processing Systems (NeurIPS 2022).

static dataset, and then optimize the decision $\mathbf{x}$ against this learned model (in reinforcement learning, the proxy model is typically the action-value function and is learned via Bellman backups, thanks to the Markovian structure). However, overestimation errors in the learned model will erroneously drive the optimizer towards low-value designs that "fool" the proxy into producing a high values [23]. To handle this, prior works have proposed to regularize the learned model to be conservative by penalizing its values on unseen designs [25, 44, 49, 17], or explicitly constraining the optimizer (e.g., by using density estimation [6, 23], referred to as "policy constraints" in offline RL [30]).

In this paper, we take a different perspective for the distributional shift challenges in offline data-driven decision-making, and formulate it as domain adaptation, where the "target" domain consists of the distribution of decisions that the optimization procedure finds, and the source domain corresponds to the training distribution. Framing the problem in this way, we derive a class of methods that regulate distributional shift at the *representation level*, and admit effective hyperparameter tuning strategies. The intuition behind our approach is that, if the internal representations of the learned model are made *invariant* to the differences between the decisions seen in the training data vs. the optimized distribution, then excessive distributional shift is naturally discouraged because excessive invariance would lead the out-of-distribution points to attain mediocre values on average. That is, as the optimized decisions deviate more from the training data, the requirement to maintain invariance will cause the model's representation to lose information, and will hence force the predicted value to become closer to the average $y$ value in the dataset. This will force the optimizer to stay within the distribution of the dataset as recall that out-of-distribution points will only appear mediocre under the learned model. Of course, the representation cannot be perfectly invariant in general, because the optimized points are different from the training data, and hence the method must strike a balance. In practice, one can instantiate this idea using an adversarial training procedure.

Our main contribution is a new class of methods for offline data-driven decision-making, derived from a novel connection with domain adaptation. In this paper, we instantiate this idea in the setting of offline data-driven model-based optimization (MBO) (i.e., offline data-driven bandits), leaving the sequential offline reinforcement learning setting to future work. As discussed above, our approach, invariant objective models (IOM), optimizes the decision – in this case, a "design" – against a learned model that admits invariant representations between optimized designs and the dataset. This enables controlling distributional shift and also provides a way to derive effective offline hyperparameter tuning strategies that we show actually work well in practice. IOM can be implemented simply by combining any supervised regression procedure with a regularizer that enforces invariance. The regularizer is implemented using any discrepancy measure between two distributions, and we instantiate IOM using the $\chi^2$-discrepancy measure

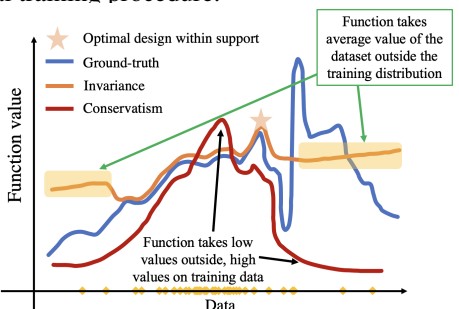

Figure 1: **Illustration showing the intuition behind IOM** on a 1D task. While conservatism (COMs) [44] pushes down the learned values on designs (decisions) not observed in the dataset and pushes up the learned values on designs (decisions) in the dataset, training with invariance induces the out-of-distribution designs (decisions) to attain objective values close to the average value in the training dataset. This discourages the optimizer from finding out-of-distribution designs (decisions).

via the least-squares generative adversarial network. This procedure trains a discriminator to discriminate between the *representations* on the training dataset and optimized designs, whereas the learned representations are trained to be as invariant as possible to fool the discriminator. We theoretically derive IOM from our domain adaptation formulation of MBO and we also use this formulation to derive tuning strategies for several common hyperparameters. Empirically, we evaluate IOM on several tasks from Design-Bench [43] and find that it outperforms the best prior methods, and additionally, admits appealing offline tuning strategies unlike the prior methods.

## 2 Preliminaries

**Offline data-driven decision-making and its variants.** The goal in offline data-driven decision-making is to produce decisions that maximize some objective function, using experience from a provided static dataset. Two instances of this problem are offline reinforcement learning (RL), and offline data-driven model-based optimization (MBO). As shown in Table 1, in offline RL, the decision is a policy $\pi$ that prescribes an action $\mathbf{a}$ for any state $\mathbf{s}$, and the objective function is the

| Quantity | Offline RL | Offline MBO |
|---|---|---|
| Decision | a policy $\pi$ mapping states to actions | a design $\mathbf{x}$ (e.g., a molecule) |
| Objective function | long-term discounted reward, $J(\pi) = \mathbb{E}_{\mathbf{a}_{0:\infty} \sim \pi}[\sum_t \gamma^t r(\mathbf{s}, \mathbf{a}_t)]$ | the objective value, $f(\mathbf{x})$ |
| Training dataset | a dataset $\mathcal{D} = \{(\tau_i, r_i)\}_{i=1}^N$ sampled from a dist. over rollouts $\tau$, $\mu(\tau)$ | a dataset $\mathcal{D} = \{(\mathbf{x}_i, y_i)\}_{i=1}^N$ sampled from the distribution of designs, $\mu(\mathbf{x})$ |
| Optimization problem | using $\mathcal{D}$ find $\arg\max_\pi \; J(\pi)$ | using $\mathcal{D}$ find $\arg\max \mathbb{E}_{\mathbf{x} \sim \mu_{\mathrm{OPT}}}[f(\mathbf{x})]$ |
| Proxy model | learn a model of $J(\pi)$, typically in the form of a value function, $\widehat{Q}(\mathbf{s}, \mathbf{a})$ | learn a model of $f$, $\widehat{f}(\mathbf{x})$ |
| A typical method | optimize the learned Q-function $\widehat{Q}$ | optimize the learned model $\widehat{f}$ |
| Distributional shift | shift between $\mu(\mathbf{a}|\mathbf{s})$ and $\pi(\mathbf{a}|\mathbf{s})$ | shift between $\mu(\mathbf{x})$ and $\mu_{\mathrm{OPT}}(\mathbf{x})$ |

Table 1: **Instantiation of our generic formulation of offline decision-making** in the context of offline RL and offline MBO. While the rest of the paper will adopt the terminology from offline MBO, the ideas we present in this paper can also be applied to offline RL.

long-term discounted reward attained by the policy $\pi$, denoted as $J(\pi)$. In offline MBO, the decision corresponds to a design $(\mathbf{x})$, and the objective function is the unknown black-box utility $(f(\mathbf{x}))$. We will utilize the notation from offline MBO in this paper for simplicity, but the ideas developed in this paper can be extended to the RL setting.

**Offline model-based optimization (MBO).** As mentioned above, the goal in offline model-based optimization (MBO) [43] is to produce designs that maximize some objective function $f(\mathbf{x})$, by only utilizing a dataset $\mathcal{D} = \{(\mathbf{x}_i, y_i)\}_{i=1}^n$ of designs $\mathbf{x}_i$ and their corresponding objective values $y_i$. No active queries to the groundtruth function are allowed. Typically offline MBO is formulated as finding the optimal design $\mathbf{x}^\star$. However, we will use a slightly more general formulation, where the goal is to optimize a *distribution* over designs $\mu(\mathbf{x})$. We will use this formulation largely for convenience of analysis, where the classic case is recovered when $\mu(\mathbf{x})$ is a Dirac delta function, though we also note that optimizing distributions over designs is often studied in settings such as synthetic biology, where the goal is to explicitly diversify the designs [5]. Thus, our goal is to find a distribution over designs $\mu_{\mathrm{OPT}}(\mathbf{x})$ using only the static dataset $\mathcal{D}$, such that it maximizes the *expected value* of $f(\mathbf{x})$ (the expected value under distribution $\mu$ is referred to as $J(\mu)$). This can be formalized as follows (Alg refers to an algorithm):

$$\mu_{\mathrm{OPT}} := \arg\max_{\mathrm{Alg}} \; \mathbb{E}_{\mathbf{x} \sim \mu(\mathbf{x})}[f(\mathbf{x})] := J(\mu) \quad \text{s.t.} \;\; \mu = \mathsf{Alg}(\mathcal{D}). \tag{1}$$

**Model-based optimization methods.** The methods we consider in this paper fit a learned model $f_\theta(\mathbf{x})$ to $(\mathbf{x}_i, y_i)$ pairs in the dataset via a standard supervised regression objective, with an additional regularizer. We will assume that $f_\theta$ is composed of two components: a representation $\phi \in \mathbb{R}^d$ that maps a given $\mathbf{x}$ into a $d$-dimensional vector, and a forward model, $f_\theta$ that converts the representational output into a scalar objective value, i.e., $f_\phi(\mathbf{x}) := f_\theta(\phi(\mathbf{x}))$. We will use the notation $f_\theta$ and $f_\phi$ interchangeably. Given a learned model $f_\theta(\phi(\mathbf{x}))$, one approach to obtain the optimized distribution $\mu_{\mathrm{OPT}}$ is by optimizing the expected value $\widehat{J}_\theta(\mu) := \mathbb{E}_{\mathbf{x} \sim \mu}[f_\theta(\phi(\mathbf{x}))]$ under the learned model. This distribution $\mu_{\mathrm{OPT}}(\mathbf{x})$ can be instantiated in many ways: while some prior work [23, 6] represents $\mu_{\mathrm{OPT}}$ via a parametric density model over $\mathcal{X}$ and optimizes this density model to obtain $\mu_{\mathrm{OPT}}$, other prior work [44] uses a non-parametric representation for $\mu_{\mathrm{OPT}}$ using a set of optimized "design particles".

In this paper, we represent $\mu_{\mathrm{OPT}}$ using a set of design particles, each of which is obtained by optimizing the learned function $f_\theta(\phi(\mathbf{x}))$ with respect to the input $\mathbf{x}$ via grad. ascent, starting from *i.i.d.* samples from the dataset. Formally:

$$\mathbf{x}_{k+1}^i \leftarrow \mathbf{x}_k^i + \eta \nabla_{\mathbf{x}} f_\theta(\phi(\mathbf{x}))|_{\mathbf{x} = \mathbf{x}_k^i}, \;\; \text{for} \;\; k \in [1, T], \;\; \mathbf{x}_0^i \sim \mathcal{D} \tag{2}$$

The optimized distribution $\mu_{\mathrm{OPT}}$ is then given by $\mu_{\mathrm{OPT}} = \delta\{\mathbf{x}_T^1, \mathbf{x}_T^2, \cdots\}$.

**The challenge of distributional shift in offline MBO** [23, 7, 9, 44]. When the learned model $f_\theta(\phi(\mathbf{x}))$ is trained via supervised regression on the offline dataset $\mathcal{D}$: $\arg\min_\theta \frac{1}{n} \sum_{i=1}^n (f_\theta(\phi(\mathbf{x}_i)) - y_i)^2$, the ERM principle states that $f_\theta(\phi(\mathbf{x}))$ will reasonably approximate $f(\mathbf{x})$ in expectation only under the distribution of the training dataset $\mathcal{D}$, i.e., $\mu_{\mathrm{data}}$, however $f_\theta(\phi(\mathbf{x}_i)) \neq y_i$ in general on points not in the training distribution. Then, when optimizing the learned function, *overestimation errors* in the value of $f_\theta(\phi(\mathbf{x}))$ will inevitably lead the optimizer to converge to a distribution $\mu_{\mathrm{OPT}}$

over designs that have a high value under the learned function(i.e., high $J_\theta(\mu_{\text{OPT}})$) but not a high ground truth value $J(\mu_{\text{OPT}})$. In general, this distribution $\mu_{\text{OPT}}$ will be different from $\mu_{\text{data}}$, since the learned function is likely to be more correct in expectation under $\mu_{\text{data}}$.

## 3 Data-Driven Offline Model-based Optimization as Domain Adaptation

In this section, we will formulate data-driven offline MBO as domain adaptation, which will allow us to derive our method, invariant objective models (IOM). Instead of attempting to simply constrain the optimizer to directly avoid distributional shift, our approach modifies the representations learned by the model $\hat{f}_\theta$ by adding an invariance regularizer during training such that there is minimal distribution shift under the learned invariant representation during optimization.

If we can ensure that the learned model $f_\theta$ is accurate on the distribution found by the optimizer, $\mu_{\text{OPT}}$, then we can ensure that we will obtain good designs. Therefore, to handle the distributional shift, we can treat $\mu_{\text{data}}$ as the "source domain" and the optimized distribution $\mu_{\text{OPT}}$ as the "target domain". Just as the goal in domain adaptation is to train on the source domain and make accurate predictions on the target domain, we will aim to train $f_\theta(\mathbf{x})$ on $\mu$ and with the goal of making more accurate predictions under $\mu_{\text{OPT}}$. Of course, learning to make accurate predictions on *any* unseen distribution is impossible for any domain adaptation method [53], but in the case of MBO, the choice of the target distribution $\mu_{\text{OPT}}$ is also made by the algorithm itself, such that invariance can be enforced *both* by changing the representation and changing the target distribution $\mu_{\text{OPT}}$.

### 3.1 Invariant Objective Models (IOM)

IOM controls distributional shift between the optimized distribution $\mu_{\text{OPT}}$ and the training distribution $\mu_{\text{data}}$ at the representational level, by regularizing the distribution of the features $\phi(\mathbf{x})$ under $\mu_{\text{OPT}}$ to match that under $\mu_{\text{data}}$. We will first present the intuition behind the method, and then formalize it by setting up a bi-level optimization problem that we will also analyze theoretically.

**Intuition:** As the optimizer deviates from the training distribution $\mu$, the invariance regularizer forces the distribution of optimized designs to resemble the distribution over representations $\phi(\mathbf{x})$ under $\mu_{\text{data}}$, i.e., $\mathbb{P}_{\mu_{\text{OPT}}}(\phi(\mathbf{x})) \approx \mathbb{P}_{\mu_{\text{data}}}(\phi(\mathbf{x}))$. When $\mu_{\text{OPT}}$ is far from the training distribution (i.e., if $D_{\text{KL}}(\mu_{\text{OPT}}, \mu_{\text{data}})$ is very high), it is easier to enforce invariance as it does not conflict with fitting $f_\theta$ on the training distribution. This representational invariance in turn causes the expected value of the learned function $f_\theta$ under $\mu_{\text{OPT}}$ to appear (roughly) equal to the average objective value in the training dataset, i.e., $\mathbb{E}_{\mathbf{x} \sim \mu_{\text{OPT}}}[f_\theta(\phi(\mathbf{x}))] \approx \mathbb{E}_{\mathbf{x} \sim \mu_{\text{data}}}[f_\theta(\phi(\mathbf{x}))]$, if $\mu_{\text{OPT}}$ is too far away from the training distribution. As a result, the expected value of the learned model $f_\theta(\phi(\mathbf{x}))$ under $\mu_{\text{OPT}}$, $J_\theta(\mu_{\text{OPT}})$, will be worse than the value of the best design within the training distribution (since the best design input sampled from $\mu_{\text{data}}$ will attain a higher objective value than the average value under $\mu_{\text{data}}$). This would act as a guard to prevent the optimizer diverging too far away from the dataset, as out-of-distribution designs no longer appear promising. Iteratively regularizing $\phi$ to enforce invariance between $\mu_{\text{OPT}}$ and $\mu$ until convergence will restrict the optimizer from selecting erroneously overestimated out-of-distribution inputs. We supplement this intuition with an illustration shown in Figure 9. We will now present a formalization of this intuition in the form of a bi-level optimization problem that aims to maximize a lower-bound on the expected objective value attained under $\mu_{\text{OPT}}$, and derive the training objectives for IOM using this bi-level optimization formulation.

However, do note that, our practical method does not aim to solve a bi-level optimization exactly due to computational infeasibility.

**Bi-level optimization for invariant objective models (IOM).** To enforce invariance at a representational level, we utilize an invariance regularizer. Denoted as $\text{disc}_\mathcal{H}(p, q)$, our invariance regularizer is a discrepancy measure between two probability distributions $p$ and $q$, w.r.t. a certain hypothesis class $\mathcal{H}$. For example, if $\mathcal{H}$ is the class of all 1-Lipschitz functions, $\text{disc}_\mathcal{H}$ is the 1-Wasserstein distance; if $\mathcal{H}$ is the class of linear functions, $\text{disc}_\mathcal{H}$ is the maximum mean discrepancy (MMD) [13] distance. In practice,

---

**Algorithm 1** IOM: Training and Optimization

1: **Input**: training data $\mathcal{D}$, number of gradient steps $T = 50$ to optimize $\mu_{\text{OPT}}$ starting from the training distribution $\mu$, training iteration $K$, batch size $n$,
2: **Initialize**: representation model $\phi_\eta(\cdot)$, learned model $\widehat{f}_\theta(\cdot)$, set of optimized design particles $\{\mathbf{x}_i^*\}_{i=1}^m$
3: **for** $k = 1 \cdots, K$ **do**,
4:     Sample $n$ training points $(\mathbf{x}_i, y_i) \sim \mathcal{D}$
5:     Run one gradient step (Equation 4) w.r.t $\theta_k$ and $\phi_k$ to obtain $\phi_{k+1}$ and $f_{\theta_{k+1}}(\phi_{k+1}(\cdot))$.
6:     Run one gradient step to optimize design particles w.r.t. $f_\theta(\phi(\mathbf{x}))$: $\mathbf{x}_i^* \leftarrow \mathbf{x}_i^* + \eta \nabla_\mathbf{x} f_{\theta_k}(\phi_k(\mathbf{x}_i^*))$.
7: **end for**

---

we implement this via a generative adversarial network as we discuss in the next subsection.

In addition to training $f_\theta$ via standard supervised regression on the labels in the training data, IOM trains the representation $\phi(\mathbf{x})$ and the model $\widehat{f}(\cdot)$ to additionally minimize the discrepancy $\mathrm{disc}_{\mathcal{H}}(\mathbb{P}_\mu(\phi(\mathbf{x})), \mathbb{P}_{\mu_{\mathrm{OPT}}}(\phi(\mathbf{x})))$ between distributions of representations under the optimized distribution, $\mu_{\mathrm{OPT}}$ and the training distribution $\mu$. This objective can be formalized as:

$$\max_{\mu_{\mathrm{OPT}}} \quad \mathbb{E}_{\widehat{\mathbf{x}} \sim \mu_{\mathrm{OPT}}} \left[ f_\theta^*(\phi^*(\widehat{\mathbf{x}})) \right]$$

$$\text{s.t. } (\phi^*, f_\theta^*) = \arg\min_{\phi, \widehat{f}} \ \frac{1}{n} \sum_{i=1}^n (\widehat{f}(\phi(\mathbf{x}_i)) - y_i)^2 + \lambda \cdot \mathrm{disc}_{\mathcal{H}}(\mathbb{P}_{\mu_{\mathrm{data}}}(\phi(\mathbf{x})), \mathbb{P}_{\mu_{\mathrm{OPT}}}(\phi(\mathbf{x}))), \quad (3)$$

where $\lambda$ denotes a weighting hyperparameter. In Section 3.2, we will describe how we can convert this bi-level optimization problem into a practical MBO algorithm that can be implemented with high capacity deep neural networks. This is followed by a theoretical analysis to formally justify the objective in Section 3.3, and a discussion of how the hyperparameter $\lambda$ can be tuned against a validation set in Section 4, without any online access to the ground truth function.

## 3.2 Optimizing the Bi-Level Problem and the Practical IOM Algorithm

In this section, we will describe how we can convert the abstract bi-level optimization problem above into an objective that can be trained practically and discuss some practical implementation details. IOM alternates between solving the bi-level optimization problem with respect to the distribution $\mu_{\mathrm{OPT}}$ and the learned model $f_\theta$ and $\phi$ independently. For training the learned model and the representation $f_\theta$ and $\phi$, IOM solves the inner optimization by running one-step of gradient descent (GD), utilizing the current snapshot of $\mu_{\mathrm{OPT}}$ (denoted as $\mu_{\mathrm{OPT}}^t$):

$$(\phi_{t+1}, \theta_{t+1}) = \mathsf{GD}_{\theta,\phi;t} \left( \frac{1}{n} \sum_{i=1}^n (f_\theta(\phi(\mathbf{x}_i)) - y_i)^2 + \lambda \cdot \mathrm{disc}_{\mathcal{H}}(\mathbb{P}_{\mu_{\mathrm{data}}}(\phi(\mathbf{x})), \mathbb{P}_{\mu_{\mathrm{OPT}}^t}(\phi(\mathbf{x}))) \right). (4)$$

Simultaneously, we update the design particles representing $\mu_{\mathrm{OPT}}^t := \delta\{\mathbf{x}_t^1, \mathbf{x}_t^2, \cdots\}$ towards optimizing the learned function $f_{\theta_{t+1}}(\phi_{t+1}(\mathbf{x}))$: $\mathbf{x}_{t+1}^i \leftarrow \mathbf{x}_t^i + \eta \nabla_{\mathbf{x}} f_{\theta_{t+1}}(\phi_{t+1}(\mathbf{x}))|_{\mathbf{x}=\mathbf{x}_t^i}, \ \forall i$. Pseudocode for IOM is shown in Algorithm 1.

**Practical implementation details:** We model the representation $\phi(\mathbf{x})$ and the learned function $f_\theta(\cdot)$ each as two-hidden layer ReLU networks with sizes 2048 and 1024, respectively. We utilize the $\chi^2$-discrepancy, which be instantiated in its variational form via a least-squares generative adversarial network (LS-GAN) [32] on the representations $\phi(\mathbf{x})$. More implementation details can be found in Appendix C.

## 3.3 Theoretical Analysis of Invariant Representation Learning for MBO

In this section, we will formally show that under certain standard assumptions, solving the bi-level optimization problem (Equation 3) enables us to find a better design than the dataset distribution $\mu_{\mathrm{data}}$. To show this, we will combine theoretical tools for analyzing offline RL algorithms [8, 25, 50] and domain adaptation [53]. We show that the expected value of the ground truth objective under any given distribution $\pi(\mathbf{x})$ of design inputs can be lower bounded in terms of the expected value under the learned function modulo some error terms. Then we will show that these error terms are controlled tightly when solving the bi-level problem, and finally appeal to uniform concentration to show that the performance of $\mu_{\mathrm{OPT}}$ is better with high probability. Our goal isn't to derive the tightest possible bound for IOM, but show that IOM can attain reasonable performance guarantees, and this domain adaptation perspective can tackle distributional shift.

**Notation and assumptions.** Following standard assumptions [29], we will analyze the setting when the learned representation lies in some space of functions, $\phi \in \Phi$, and the learned function $f_\theta$ lies in $f_\theta \in \mathcal{F}$. We will consider that the optimized distribution $\mu_{\mathrm{OPT}}$ belongs to a function class, $\mu_{\mathrm{OPT}} \in \Pi$. Our analysis will require that the ground truth model $f$ is approximately realizable under some representation $\phi \in \Phi$ and for some $g \in \mathcal{F}$. That is, we assume:

**Assumption 3.1** ($\varepsilon$−realizability). For any distribution $\pi$ over designs, there exists a representation $\phi \in \Phi$ and a $g \in \mathcal{F}$ such that $\min_g \max_{\pi \in \Pi} \mathbb{E}_{\mathbf{x} \sim \pi} [\|f(\mathbf{x}) - g(\phi(\mathbf{x}))\|] \leq \varepsilon_{\mathcal{F}, \Phi}$.

In addition, we assume that $\forall f \in \mathcal{F}, \|f\|_\infty < \infty$ and is Lipschitz continuous with constant $C$: $\|f\|_{\mathrm{L}} \leq C$. This is true for most parameteric function classes, such as neural networks. Then, we can lower bound the average objective value under any distribution in terms of the invariance:

**Proposition 3.2** ((Informal) Lower-bounding the ground truth value under $\pi$). *Under Assumption 3.1 and the Lipschitz continuity on the function $f_\theta$, the ground truth objective for any given $\pi \in \Pi$ can be lower bounded in terms of the learned objective model $f_\theta \in \mathcal{F}$ and representation $\phi \in \Phi$, with high probability $\geq 1 - \delta$ that:*

$$J(\pi) - J_\theta(\pi) \geq \underbrace{J(\mu_{data}) - J_\theta(\mu_{data})}_{(\blacksquare)} - \underbrace{C_\mathcal{F} \cdot disc_\mathcal{H}(\mathbb{P}_{\mu_{data}}(\phi(\mathbf{x})), \mathbb{P}_\pi(\phi(\mathbf{x})))}_{(*)} - \varepsilon_{\mathcal{F}, \Phi} - \varepsilon_{stat},$$

*where $C_\mathcal{F}$ is a uniform constant depending on the function class $\mathcal{F}$ and $\varepsilon_{stat}$ refers to statistical error that decays inversely with the dataset size, $|\mathcal{D}|$.*

A proof for Proposition 3.2 is in Appendix A. The $\blacksquare$ term can further be lower bounded using a standard concentration error bound for ERM, and can be controlled if we train $f_\theta(\phi(\cdot))$ minimize prediction error against ground truth values of $f$. Now we utilize Proposition 3.2 and a uniform concentration argument to lower-bound the value under the optimized distribution $\mu_{\text{OPT}}$.

**Proposition 3.3** ((Informal) Performance guarantee for IOM). *Under Assumption 3.1, the expected value of the ground truth objective under $\mu_{\text{OPT}}$, $J(\mu_{\text{OPT}})$ is lower bounded by:*

$$J(\mu_{\text{OPT}}) \gtrsim J(\mu_{data}) - \mathcal{O}\left(\sqrt{\frac{\log \frac{|\mathcal{F}||\Phi||\Pi|}{\delta}}{|\mathcal{D}|}} + \frac{\log \frac{|\mathcal{F}||\Phi||\Pi|}{\delta}}{|\mathcal{D}|}\right) + \underbrace{J_\theta(\mu_{\text{OPT}}) - J_\theta(\mu_{data})}_{(\circ)} - (*). \quad (5)$$

A proof of Proposition 3.3 can be found in Appendix A. This proposition implies that as long as the optimizer finds designs that improve the learned objective function $f_\theta \in \mathcal{F}$, such that $(\circ)$ is positive, and the discrepancy term $(\star)$ is minimized, then the distribution over designs found by IOM, $\mu_{\text{OPT}}$, will compete favorably against the distributions of designs in the dataset. Of course, these bounds present the worst-case guarantees for IOM, but as we will show in our experiments, IOM does clearly improve over the best designs in the offline dataset. In addition, this theoretical analysis also motivates the design of our scheme for performing hyperparameter tuning and model selection. This scheme will be based directly on the result from Proposition 3.2. We will discuss this connection when discussing tuning in Section 4.

Finally, we would like to highlight how invariance leads to "average" values if $\mu_{\text{OPT}}$ deviates too far away from the data distribution $\mu$. If $\mu_{\text{OPT}}$ deviates too far, such that $disc_\mathcal{H}(\mathbb{P}_{\mu_{data}}(\phi), \mathbb{P}_{\mu_{\text{OPT}}}(\phi)) \leq \varepsilon'$, the learned function is bounded too close to the average learned value on the data i.e., $|\widehat{J}(\mu_{\text{OPT}}) - \widehat{J}(\mu_{data})| \leq C' \cdot \varepsilon'$. Thus, the learned function simply reverts to "predicting the mean objective value" of the training distribution as $\varepsilon'$ decreases, thereby losing any information about the identity of the optimized distribution, when the optimizer goes too far out of $\mu$.

# 4 Offline Workflow and Tuning for IOM

A central challenge in offline model-based optimization is that of offline workflow and hyperparameter tuning: not only must an algorithm be able to produce designs using a static dataset, but it must also prescribe a *fully offline* workflow for tuning hyperparameters so that the produced designs are as good as possible. This includes both explicit hyperparameters (such as the coefficient of conservatism or invariance, $\lambda$ for IOM) and implicit hyperparameters (such as the early stopping point). The primary challenge stems from the fact that unlike supervised learning, the final performance in MBO is determined by the optimized distribution $\mu_{\text{OPT}}$, which is different from the training distribution, $\mu_{\text{data}}$, and we must reason about this performance fully offline.

In this section, we discuss how IOM admits a particularly convenient and *fully offline* hyperparameter tuning procedure that works well empirically. The goal of our tuning problem is to find a tuple consisting of the learned representation $\phi^\lambda$, the learned model, $f_\theta^\lambda$, and the optimized distribution $\mu_{\text{OPT}}^\lambda$, out of a given set of tuples obtained for different values of $\lambda$. In other words, we wish to find the $\lambda$ from the set of models $M_\lambda := (f_\theta^\lambda, \phi^\lambda, \mu_{\mu_{\text{OPT}}}^\lambda)$, such that the corresponding $\mu_{\text{OPT}}^\lambda$ attains the highest value under the ground truth objective. Crucially note that this must be done fully offline.

**Our key idea:** The primary question we wish to answer is: how can we identify if a given tuple $(f_\theta^\lambda, \phi^\lambda, \mu_{\text{OPT}}^\lambda)$ leads to good performance? In order for a tuple trained via IOM to perform well, we need two primary conditions: **(1)** it should be nearly invariant, and as a result, robust to distributional

shift, and (2) the learned $f_\theta^\lambda$ must attain good generalization performance within the training distribution. Intuitively (1) guards us against distributional shift, and (2) enables us to find as good of a learned model on the dataset, that does not overfit or underfit as a result of the invariance regularizer. This forms the basis of our tuning method.

**Offline tuning of IOM:** To instantiate the idea practically, we first run IOM with a set of hyperparameters $\Lambda = \{\lambda_1, \lambda_2, \cdots\}$. Then for each run, we record the validation in-distribution error and the value of the invariance regularizer on a validation set. We then perform tuning in two steps: first, we filter all $\lambda$ values that attain near-perfect invariance under a user-specified margin $\varepsilon$, i.e., $\text{disc}_{\mathcal{H}}(\mathbb{P}_{\mu_{\text{data}}}(\phi^\lambda(\mathbf{x})), \mathbb{P}_{\mu_{\text{OPT}}^\omega}(\phi^\lambda(\mathbf{x}))) \leq \epsilon$, and are hence guaranteed to be robust to distributional shift. Second, we now pick models that attain good performance within the training distribution by selecting $\lambda$ values that attain the smallest validation prediction error: $(f_\theta(\phi(\mathbf{x})) - y)^2$, in addition to picking the early stopping point based on the smallest validation error. This enables us to pick $\lambda$ that gives rise to a generalizing model $f_\theta^\lambda, \phi^\lambda$, while being most robust to distributional shift. This still leaves a user-defined margin $\varepsilon$ as a free parameter, but we find that it is comparatively insensitive and can be selected heuristically, as we show in our experiments in Section 5.2.

Formally, this procedure corresponds to solving the following optimization:

$$\lambda^* := \arg\min_{\lambda \in \Lambda} \ \frac{1}{n} \sum_{i=1}^{n_{\text{val}}} (f_\theta^\lambda(\phi^\lambda(\mathbf{x}_i)) - y_i)^2 \ \text{ s.t. } \ \text{disc}_{\mathcal{H}}(\mathbb{P}_{\mu_{\text{val}}}(\phi^\lambda(\mathbf{x})), \mathbb{P}_{\mu_{\text{OPT}}^\lambda}(\phi^\lambda(\mathbf{x}))) \leq \epsilon \quad (6)$$

Finally, for a more formal explanation of the soundness of this strategy, we note that applying Equation 6 provides a tight lower bound on the ground truth objective value in Proposition 3.2. Up to statistical error (which decays as $|\mathcal{D}|$ increases), the prediction error on a held-out validation set estimates (■) in Proposition 3.2, and a smaller discrepancy (i.e., $\text{disc}_{\mathcal{H}} \leq \varepsilon$) controls the ($\star$) term in Prop. 3.2. $\varepsilon_{\text{stat}}$ is statistical error, and $\varepsilon_{\mathcal{F}, \Phi}$ only depends on the function classes involved ($\mathcal{F}, \Phi$). This tuning $\lambda$ per Equation 6 can be justified as aiming to maximize a lower bound on $J(\mu_{\text{OPT}})$.

The practical implementation is in in Appendix C.3.

In Section 5, we validate this offline tuning strategy, both against other alternative offline strategies for tuning IOM as well as offline strategies for tuning prior methods (COMs [44] and gradient descent [43]). We find that our offline tuning strategy performs comparably to full online tuning without needing any online queries, whereas other strategies for IOM and other methods, lead to significantly poor performance compared to online tuning.

## 5 Experimental Evaluation

The goal of our empirical evaluation is to answer the following questions: (1) Does IOM outperform prior state-of-the-art methods for data-driven model-based optimization? (2) Is our offline tuning strategy for IOM better than other offline tuning strategies for IOM or other prior methods? (3) How sensitive is IOM to the choice of the hyperparameter $\lambda$? In this section, we will answer these questions via a comparative evaluation of IOM on several continuous data-driven design tasks from the standard tasks from the Design-Bench suite [43].

### 5.1 Empirical Evaluation on Benchmark Tasks

We compare IOM to a variety of prior methods for model-based optimization: CbAS [6], Autofocused CbAS [9] and MINs [23] that utilize generative models for constraining distributional shift; COMs [44] and RoMA [49] train robust models of the objective function by via conservatism and smoothness, respectively; gradient-free optimization methods such as REINFORCE [46], CMA-ES [15] and conventional Bayesian optimization BO-qEI [47]. We also compare the standard approach of learning a naïve model of the objective function, which is then optimized via gradient ascent. To better understand the benefits of enforcing invariance, we also compare IOM to IOM-C, which additionally applies a conservatism term on top of invariance that pushes up the learned values of designs in the dataset. Additionally, we add a prior method that optimizes designs against a lower-confidence bound estimate computed using a Gaussian process posterior in Appendix C.4.

**Tasks.** We evaluate on four tasks with continuous-valued input space from the Design-Bench [43] benchmark for offline model-based optimization (MBO). More details about these tasks can be found in Appendix C.

**Evaluation protocol.** Following prior work [43, 44, 49, 10], for each method, we query the learned model to obtain the top performing $N$ points, i.e., $\mathbf{x}_1, \mathbf{x}_2, \cdots, \mathbf{x}_N$. We report the maximum objective

value among these $N$ samples. This protocol reasonably reflects a real-world design process, where a number of computationally produced designs are tested and the best one is used for deployment.

**Hyperparameter tuning.** For fair comparison against prior methods studied in Trabucco et al. [43], we follow the uniform tuning strategy for reporting results in this section: we run IOM on each task for a given set of $\lambda$ values, and pick a single value of $\lambda$ across every task. Note that this does require access to online evaluation, but prior works do use the same protocol. We will evaluate the efficacy of our offline tuning approach in Section 5.2 extensively.

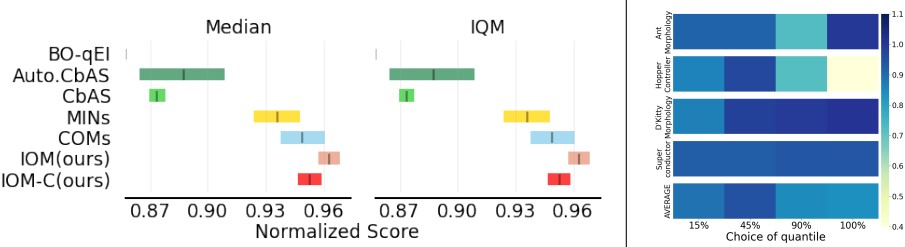

Figure 2: **Left**: Median and IQM [1] (with 95% Stratified Bootstrap CIs) for the aggregated normalized score on Design-bench. IOM improves over the prior methods, including those which use explicit conservatism although IOM does not. As we could not find the individual runs for RoMA [49], we report the mean and standard deviation for RoMA (copied from the results in Yu et al. [49]), along with more detailed results for other baselines in Appendix C.4. **Right**: The offline tuning performance matrix under different quantiles to filter the models based on discriminator loss, compared with the oracle uniform tuning, which shows our tuning strategy is robust to this hyper-parameter.

**Results.** Following the convention proposed by Trabucco et al. [43], we evaluate all methods in terms of normalized score (higher is better). In Figure 2 (left), we show the median and interquantile mean (IQM) [1] for the aggregated scores across all tasks. Due to space limits, we only report the scores for the top-performing baselines (See Appendix C for the results for all baselines). IOM significantly outperforms all of the prior methods, including state-of-the-art methods that employ conservatism, such as COMs [44]. This indicates that invariance alone (IOM) can effectively handle distribution shifts and avoid over-estimation for out-of-distribution actions. Additionally, a comparison between the performance of IOM and IOM with conservatism (IOM-C) suggests that adding conservatism does not improve performance further. This further corroborates the fact that invariance is effective in preventing the optimizer from going out of distribution and conservatism is not needed. That said, we would like to clarify that this result does not mean that invariance will always be better than conservatism on any given offline MBO problem. In order to understand why IOM outperforms COMs on these tasks, we examine the in-distribution validation error of IOM and COMs in Appendix E and find that IOM learns less distorted functions that generalize better.

## 5.2 Analysis of the Offline Workflow for Tuning IOM

We will now present experiments to understand the effectiveness of our offline tuning scheme (Section 4) for tuning IOM.

It is important to highlight that, in general, offline tuning presents significant challenges for MBO. Prior works in bandits and reinforcement learning [52] have suggested that offline tuning should not work at all and, to the best of our knowledge, no prior work in offline MBO has proposed a principled and effective offline tuning method. In this section, we empirically validate our tuning strategy in comparison to other potential strategies, and compare results from our approach to an oracle strategy that uses online evaluation of the true function.

**Comparisons.** We compare our offline tuning strategy to various offline strategies: (1). only tunes IOM based on validation-set prediction error, analogous to supervised learning (called "prediction error only"); (2). a strategy that tunes IOM based on only the value of the invariance regularizer (called "invariance only"); (3). a method that utilizes our tuning strategy for picking $\lambda$, but not for picking the checkpoint ("uniform checkpoint selection") and two offline tuning strategies for tuning other prior methods: (a) tuning COMs [44] using validation-set prediction error (called "prediction error only"), conservatism loss (called "conservatism loss only") and both prediction error and conservatism loss, one by one (called "two-step tuning"); and (b) tuning an objective model trained via gradient ascent [43] using validation MSE error ("gradient ascent"). We report the drop in performance relative to the corresponding oracle online strategy that actually computes the "groundtruth" values of optimized designs in the simulator to choose hyperparameters in hindsight, i.e., the relative drop in

performance of the method (IOM, COM, grad. ascent) when tuned offline using the particular strategy compared to oracle tuning for the same method.

**Results.** We present our results in Figure 3 (right) comparing the efficacy of various tuning schemes across various methods averaged over all tasks. While offline tuning does lead to a reduction in performance compared to oracle online tuning, we find that the drop is smallest for our offline tuning strategy applied to IOM. All other tuning strategies based on either only prediction error or the invariance regularizer lead to worse performance. This indicates that our tuning strategy is effective in tuning IOM. Furthermore, offline tuning strategies that utilize a validation set with prior methods, such as COMs and gradient ascent, also lead to a significant decrease in performance relative to the oracle. This is quite appealing as it suggests that IOM is more amenable to offline tuning strategies relative to prior offline MBO algorithms. Even if these prior methods can perform well when provided access to online evaluations, they might start to fail in real-world MBO problems when one needs to tune them fully offline. We discuss computational complexity of tuning the methods in Appendix E.

**Ablation analyses.** Next we aim to study the sensitivity of our offline tuning strategy with respect to the non-rigorous, user-specified parameter $\varepsilon$ in Equation 6 and checkpoint selection based on the validation prediction error. Figure 2 (right) shows the offline tuning performance heatmap for all tasks (along their aggregated performance in the bottom row) under different choice of quantiles (where quantile $x\%$ means the top $x\%$ of the models according to the value of the invariance regularizer, and the given value of $\epsilon$ were chosen). The color in each position of the matrix corresponds to the ratio of performance of our offline tuning strategy for a given quantile value, compared to that of uniform oracle tuning. As shown in the bottom row, aggregated over all tasks, we can see our offline selection strategy is quite robust to the hyperparameter $\epsilon$. Figure 3 (left) evaluates the efficacy of our checkpoint selection strategy, we plot the objective value $J(\mu_{\mathrm{OPT}})$ attained as a function of the number of training epochs (i.e., checkpoints), alongside the values for the prediction MSE in the validation set for the corresponding epochs. The checkpoints selected by our early stopping scheme are shown via the vertical line. Observe that indeed the checkpoint for the lowest validation MSE corresponds to a rather good optimized distribution $\mu_{\mathrm{OPT}}$ for different tasks, no matter the optimal training epochs happen at the beginning or near the end, which validates our early stopping strategy. More detailed results showing the efficacy of early stopping for both IOM can be found in Appendix C.4.

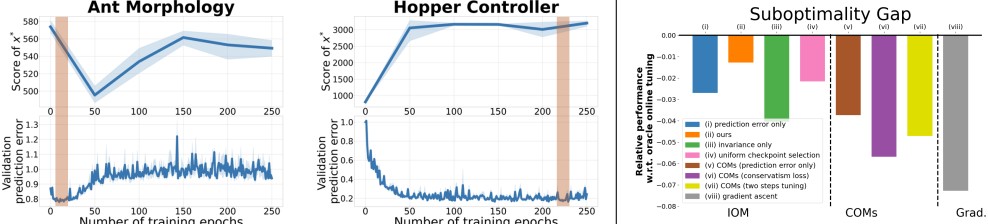

Figure 3: **Left**: Visualizing our checkpoint selection scheme on two tasks, and no matter the best checkpoint happens at the beginning or end of the training, our strategy based on validation in-distribution MSE could capture the best performance. **Right**: Offline tuning vs oracle uniform online tuning for IOM, COMs and Gradient-ascent. Among all the methods, IOM achieves the smallest regret for offline tuning, and our specific tuning strategy works best compared with other tuning strategies for IOM.

## 6    Discussion

In this work we study offline data-driven model based optimization (MBO), where the goal is to produce optimized designs, purely using a static dataset. We proposed to view offline MBO through the lens of domain adaptation, and proposed a method that trains a model of the objective with an additional regularizer to promote invariance between the representations learned by the model in expectation under the training data distribution and the distribution of optimized designs. We show that this leads to a practically effective data-driven optimization method, and an appealing workflow for selecting the hyperparameters with offline data. Our method outperforms prior approaches, which illustrates the practical benefits of invariance. We also validate the effectiveness of the accompanying tuning strategy. While we evaluate our method on MBO problems, our approach can in principle also be extended to contextual bandits and even to, sequential reinforcement learning. But we admit that there might be some limitations: **(1)** fully data-driven offline optimization may on its own be insufficient (due to limitations of staying close to the dataset, hardness of tuning) and inevitably will require access to online evaluation. **(2)** a full understanding of theoretical conditions when invariance can outperform conservatism is important, but out of the scope of this work.

## Acknowledgements

We thank Brandon Trabucco and Xinyang Geng for help in setting up the Design bench and COMs codebases. We thank anonymous NeurIPS reviewers, members of the RAIL lab at UC Berkeley and Amy Lu for informative discussions, suggestions and feedback on an early version of this paper. This research was supported by the Office of Naval Research, C3.ai, and Schmidt Futures, and compute resources from Google cloud. AK is supported by the Apple Scholars in AI/ML PhD fellowship.

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
