# OpenReview forum: "Data-Driven Offline Decision-Making via Invariant Representation Learning"
_NeurIPS.cc/2022/Conference — NeurIPS 2022 Accept_

### Official Review · Reviewer_VghW · 2022-07-05

**Rating:** 5
**Confidence:** 2
**Soundness:** 3 good
**Presentation:** 2 fair
**Contribution:** 3 good

**Summary:**

The paper considers the problem of model-based black-box optimization given a static dataset. The main concern is not to report a design that is influenced by model overestimations. The authors propose to formulate this problem as domain adaptation and propose an algorithm IOM that formulates the problem as bi-level optimization and enforces invariance at the representation level by using a specifically designed regularizer. Experimental results show the comparison against other methods for model-based optimization.


**Questions:**

Please see the previous section.

**Limitations:**

The authors only stated one limitation that is again very vague in the conclusion.

**Strengths And Weaknesses:**

I had a hard time reading this paper and this is the primary reason for not being able to grasp the main contributions. Overall, I think that the text can be significantly improved. Even the sections that were supposed to convey intuition were confusing to me. For example, the “Intuition (Line 156)” section did not convey intuition and confused me even more. I did not understand sentences such as:
* Such an invariance can be enforced when $\mu_{OPT}$ is sufficiently far away from the training distribution.
* .. if $\mu_{OPT}$ is too far away from the training distribution.
* This would pull the optimizer closer to the training distribution.

The abstract should do a better job of highlighting the main contributions better. IOM is used in the abstract, but it is unclear what IOM stands for.

Not sure what the main message of Figure 1 is. I think it does not convey intuition well (the last sentence is unclear).
Things should also be more formal. $\mu_{OPT}$ and $J(\mu)$ have the same definition in Eq. 1. It would also be helpful if the distributional shift that the paper talks about is mathematically formalized.

I was also unable to comprehend the theoretical result. If the circle is positive and the star is minimized, then mu_{OPT} is better than mu_{data}. I’m not sure what the utility of such a result is. Again, why do we care about mu_{data} when optimizing, and not only consider the best point/value given the noiseless setting.

Overall, I feel that the entire method is overly complicated. I am not convinced that we need to solve the bi-level optimization problem (which is known to be notoriously hard). While I understand why one might be interested in obtaining “soft” allocation (i.e., distribution over solutions), I do not understand why we need to look at the entire training data distribution and not the best point/value within the given training data.

The first experiment is also unclear to me. Do you compare it against the best point in the training dataset? How can CMA-ES and REINFORCE be conventional Bayesian Optimization methods?
I’d suggest looking at more relevant arXiv:2002.09038 and similar model-based (BO) robust works. Another question is how fitting a GP to the training data, defining confidence bounds, and outputting the design that corresponds to the max lower confidence bound performs in this experiment?

Some minor comments:
– 41: Typo “invariance”
– 127: Is this Dirac $\delta$?

---

> ### Author Response · Authors · 2022-08-01
> **Author Response (Part 1 of 2): Clarified experiments, added LCB + GP baseline, bi-level optimization**
>
> We thank the reviewer for their detailed comments. To address the reviewer’s concerns regarding clarity, we have updated the intuition for our method in the paper (**Lines 158 - 175**). We have also supplemented this intuition with a new flowchart figure **Figure 10 (Page 25)** that we believe will greatly simplify the exposition. We have also updated the paper to discuss more concrete limitations of this paper (Section 7), updated the abstract, corrected the typos and notation overloads, and added the additional experiments as we discuss in detail below. The revisions to the paper are shown in $\textcolor{red}{red}$. **We are happy to make further revisions to improve clarity and address any remaining concerns, please let us know if that would be helpful.**
>
> ____
>
> > **The first experiment is also unclear to me. Do you compare it against the best point in the training dataset?**
>
> The comparison to the best point in the training dataset is already present in our comparisons in **Table 2** in Appendix A.3.1 and yes, we find that IOM outperforms the best point in the training dataset (compare to $\mathcal{D}(\text{best})$).  Additionally the set of prior methods we compare corresponds to the set of all methods evaluated in the Design bench benchmark, which provides standardized evaluation scores, and several prior ICML, NeurIPS and ICLR papers evaluate their offline MBO methods on this benchmark (Yu et al. NeurIPS 2021, Trabucco et al. ICML 2021, 2022).
>
> ___
>
> > **Another question is how fitting a GP to the training data, defining confidence bounds, and outputting the design that corresponds to the max lower confidence bound performs in this experiment?**
>
> We already have a baseline (Grad (min)) that optimizes the design against the lower-confidence estimate obtained from a neural network ensemble posterior and not a GP posterior, since GPs are typically not as scalable when dealing with high dimensions (HopperController has 5000 dimensions). While IOM already outperforms this Grad (min) baseline (see Table 2 in Appendix),  per the reviewer’s suggestion, we have also now added the suggested GP baseline. Observe in the new results in **Table 2** in Appendix A.3.1 that IOM outperforms this GP baseline in three tasks we benchmark on (due to limited time, we only got the results for three tasks). We would also like to note that the methods we compare correspond to the complete set of baselines from the Design bench benchmark (Trabucco et al. ICML 2022) from where we borrow these tasks.
>
> ___
>
> > **How can CMA-ES and REINFORCE be conventional Bayesian Optimization methods?**
>
> Our apologies for listing them as BO methods. We have updated the text to clarify that these are not Bayesian Optimization methods, however note that the BO-qEI method we compare to is a Bayesian optimization method (Wilson et al. BayesOpt 2017).
>
> ___
>
> > **I’d suggest looking at more relevant arXiv:2002.09038 and similar model-based (BO) robust works.**
>
> Thanks for pointing out these related works, we have now cited and discussed the above mentioned paper and other works in the area of robust BO from https://github.com/ilijabogunovic/Robust-BO. However, we believe that our problem setting is distinct from arxiv:2002.09038: while this prior work studies robustness across contexts in model-based optimization problems, in our evaluation we focus on non-contextual MBO problems, and hence these prior methods are not applicable. Even our method, IOM, only attempts to handle adversarial designs that fool the optimizer, and does not seek to be robust to the context distribution, which is an orthogonal aspect.
>
> ___
>
> > **bilevel optimization is notoriously hard; method is complicated**
>
> We would like to clarify that we do not solve the exact bi-level optimization in practice, but it is only used as a formalism for our analysis, similar to prior works in the domain of reinforcement learning. We have updated the discussion in Section 4.1 to reflect this, and would clarify that in order to attain good results in practice, IOM simply needs a model of the objective with an invariance regularizer, which is then optimized via naive gradient descent. In practice, adding this regularizer is no more complicated than standard invariance losses in domain adaptation (e.g., DANNs).

---

> > ### Author Response · Authors · 2022-08-01
> > **Author Response (Part 2 of 2): Clarification regarding theoretical result and distributional shift**
> >
> > > **Theoretical result does not consider the best point/value given the noiseless setting**
> >
> > We initially expressed the result as an improvement over the average value in the dataset following prior works in offline reinforcement learning, that care about improvement over the average behavior policy (e.g., safe-policy improvement in SPIBB (Laroche et al. ICML 2020), CQL (Kumar et al. 2020), Bellman-consistent pessimism (Xie et al. 2021)).
> >
> > However, to address this concern, we have added an extension of the theoretical result  in **Appendix A.1.4**, which lower-bounds the evaluation performance $J(\mu_\mathrm{OPT})$ with respect to the average objective value of any re-weighting of the training data distribution. This subsumes the case of bounding the performance gap with respect to the best design in the dataset, and we believe this should resolve the reviewer's concern.
> >
> > ___
> >
> > > **formalizing distributional shift**
> >
> > We already have a paragraph discussing the challenge of distributional shift in offline model-based optimization in Lines 129-137. We have now utilized formal notations in that paragraph to make it clearer and briefly summarize this below. Does this address the concern?
> >
> > **Where does the distribution shift arise?** The distributional shift is between the distribution of samples found by the optimizer ($\mu_\mathrm{OPT}$) and the data distribution ($\mu_\text{data}$).
> >
> > **Why does the distribution shift arise?** When a naive model of the objective function ${f}_\theta(\mathbf{x})$ is trained via ERM,  ${f}_\theta(\mathbf{x}) \neq f(\mathbf{x})$ on $\mathbf{x}$ s.t. $\mu_\text{data}(\mathbf{x}) = \text{small}$. This causes the optimizer to find a design distribution $\mu_\text{OPT}$ which is different from $\mu_\text{data}$, and where the expected learned value is overestimated.
> >
> > ___
> >
> >
> > **Please let us know if these answers and updates to the paper address your concerns.**

---

> ### Author Response · Authors · 2022-08-05
> **Request for discussion**
>
> Dear Reviewer VghW,
>
> We were wondering if you have gotten a chance to go through our responses and the revisions to the paper, and if these revisions and responses address your concerns regarding the paper. We are happy to address any remaining concerns and would really appreciate it if you engage in a discussion with us.
>
> Thank you so much!

---

> > ### Author Response · Authors · 2022-08-07
> > **Discussion**
> >
> > Dear Reviewer VghW,
> >
> > There are only 2 more days remaining in the author-reviewer discussion period -- we were wondering if you have gotten a chance to go through our responses and the revisions to the paper. We are happy to address any remaining concerns and would really appreciate it if you engage in a discussion with us.
> >
> > Thank you so much!

---

> > > ### Comment · Reviewer_VghW · 2022-08-08
> > > **Response**
> > >
> > > The author's response and revisions have addressed the majority of my concerns/questions and hence I updated my rating.

---

### Official Review · Reviewer_1x47 · 2022-07-10

**Rating:** 5
**Confidence:** 5
**Soundness:** 3 good
**Presentation:** 3 good
**Contribution:** 3 good

**Summary:**

This paper proposes a model-based optimization method, which handles the distribution shift challenges through the perspective of domain adaptation. The key idea is to regularize the learned model by treating the training data as the “source domain” and the optimized designs as the “target domain”. The invariance optimizer forces the distribution shift between source and target to be minimal at the representation level, which ensures the out-of-distribution points will not be erroneously overestimated. Overall, the idea of this work is novel to me and the intuition behind is convincing to some extent. Although I didn’t check all details, the proposed bi-level optimization problem and the corresponding practical algorithm appear to be technically sound.

**Questions:**

1. The distribution shift in MBO is not clearly defined or illustrated in this paper. In domain adaptation, the source and target domains have different data distributions but share the same feature space. In this paper, the target optimized designs are obtained by optimizing the learned model with respect to the inputs via gradient ascent, similar to adversarial samples. My main concern is whether the adversarial samples equivalent to the case in domain adaptation?
2. In section 6.1-Evaluation protocol, how do you test or deploy the produced designs?


**Limitations:**

1. The bi-level optimization objective is not convincing to me. Why do authors formalize IOM training as a bi-level optimization problem with respect to the optimal distribution and the learned model parameters simultaneously? The standard approach of learning a naïve model, which is then optimized via gradient ascent with a invariant regularization term also makes sense to me. Explanations on the necessity of bi-level optimization are welcome.
2. The author claimed that invariance can effectively handle distribution and conservatism is not necessary. Does it mean IOM is consistently better than conservatism? Any insightful explanations? I suppose that the conclusion needs more examples to support.
3. The offline tuning strategy needs multiple runs of IOM with a set of hyperparameters, which may be time-consuming. The computational efficiency should be considered.
4. In section 6.2, more details on oracle turning should be added. Besides, for suboptimality gap in Figure 3, the comparisons with COMs are not fair, since it only considers two naïve case for COMs, more tuning strategies for COMs needs to be evaluated.


**Strengths And Weaknesses:**

Strength points:
1. This paper contributes some new ideas. The key idea is formulating offline model-based optimization as domain adaptation, which may have a broad impact.
2. This paper is clearly written and well organized.
3. The practical implementation of IOM algorithm and the offline tuning is detailed.

Weaknesses:
Please refer to Limitations.

---

> ### Author Response · Authors · 2022-08-01
> **Author Response (Part 1 of 2): Domain Adaptation, Necessity of bi-level optimization**
>
> We thank the reviewer for their detailed feedback. We have now revised the discussion on distributional shift in Lines 129-137 to more clearly discuss the distributional shift problem in offline MBO. We also note that this problem has also been studied in prior papers published at ICML, NeurIPS and ICLR in recent years (Trabucco et al. ICML 2021, 2022; Yu et al. NeurIPS 2021, Kumar et al. NeurIPS 2020; Fannjiang et al. NeurIPS 2020). The revisions to the paper are shown in $\textcolor{red}{red}$.
>
> We now answer individual questions below:
>
> > **My main concern is whether the adversarial samples equivalent to the case in domain adaptation**
>
> It is definitely not obvious that domain adaptation methods address this issue -- indeed, the whole point of our paper is to show this. Hence, our theoretical results show that invariance in theory addresses the "adversarial samples," and our empirical results show that this technique works well in practice. In domain adaptation, we are provided with a source and test distribution, where the goal is to achieve good performance under the test distribution, while training on the source distribution. As we discuss in detail in the paper in Lines 144-152, offline MBO also presents two distributions – the data distribution which we train on, and the distribution of the learned optimized designs. The goal is also similar to domain adaptation: train on the data distribution, but attain good performance on the distribution of optimized designs. However, unlike standard domain adaptation, where the target distribution is fixed and given, in MBO, the optimization procedure controls the target distribution. Nevertheless, we show that despite this difference, techniques from domain adaptation can be effective in offline MBO, both in theory (Section 4) and in practice (Section 6).
>
> ___
>
>
> > **Necessity of bilevel optimization**
>
> Our practical IOM algorithm does not implement the exact bi-level optimization; we only use bi-level optimization to motivate and formulate the problem,  and we have clarified this in the revised paper now. In practice, our implementation is similar to prior methods such as COMs, where we add an extra (invariance) regularizer while training an naive objective model via supervised learning. Thus, the implementation is no more complex than other invariance regularizers (e.g., DANNs, etc.).
>
> For our theoretical analysis, we use a bi-level optimization formalism following analyses of pessimistic offline reinforcement learning methods including actor-critic methods (see for example, Cheng et al. ICML 2022 or Kumar et al. NeurIPS 2020). We have now clarified in the paper that bi-level optimization is a formalism following prior work, and it may be replaced with other analysis tools. Our practical approach doesn’t implement the exact bi-level optimization, and is no more complex than other invariance regularizers used in domain adaptation.
>
> ___
>
>
> > **How do you test or deploy the proposed designs?**
>
> In our experiments, we exactly follow the experimental protocol from the Design Bench benchmark (Trabucco et al. ICML 2022), which does provide a simulated evaluator to be used for evaluating proposed designs only. More details regarding the simulator and evaluation method specific to each task can be found in Appendix A.2 in our paper. Although this evaluation protocol has clear shortcomings, it has been used in a number of prior works, and felt it best to follow this protocol (Fu et al. ICLR 2021, Yu et al. NeurIPS 2021, Trabucco et al. ICML 2021).

---

> > ### Author Response · Authors · 2022-08-01
> > **Author Response (Part 2 of 2): invariance vs conservatism, questioins related to tuning**
> >
> > > **Does it mean IOM is consistently better than conservatism? Any insightful explanations?**
> >
> > We have toned down the claim that IOM would _always_ be better than conservatism and revised it to indicate that this conclusion holds in the domains we test on. We have now added some empirical analysis in **Figures 8 & 9** in the Appendix where we find that it is considerably harder to minimize the training prediction error (i.e., error in accurately predicting the ground-truth value of a given design $\mathbf{x}$) with COMs as opposed to IOM. Moreover, COMs learn more non-smooth functions that IOM as the norm of the gradient with respect to the input, $\mathbb{E}_{\mathbf{x} \sim \mathcal{D}}[||\nabla_x f_\theta(\mathbf{x})||_2^2]$ is higher for COMs (Figure 9).
> >
> > We suspect that this is because conservative training can distort the objective function more than IOM. Since conservatism simply pushes down the objective value on out-of-distribution points to be as small as possible, it encourages the learned model to be non-smooth. On the other hand, IOM does not force the learned objective model to be non-smooth. Of course, better training error or smoothness of the learned function may not necessarily translate to better performance in every offline MBO problem, but we believe that a more smooth model and better generalization could be helpful in practical problems. As is always the case when comparing to prior methods, the results are specific to the particular domains that are tested, and it is generally impossible to conclusively prove that one method is always better than another.
> >
> > We have now added this discussion to Appendix A.6 in the revised paper.
> >
> > ___
> >
> > > **more tuning strategies for COMs needs to be evaluated; more details on oracle tuning**
> >
> > Oracle online tuning in Section 6.2 refers to tuning hyperparameters by directly using _groundtruth_ values of the optimized designs, obtained through the simulator. That is, we train IOM (or any other method) for the set of hyperparameters, evaluate the resulting designs for each hyperparameter under the simulator, and then pick the one that attains the highest ground-truth value. This scheme is referred to as “oracle”, since in practical problems we would not have access to evaluating produced designs through the simulator for every hyperparameter and we must do offline tuning.
> >
> > We would like to note that the COMs paper does not provide any strategy for tuning that can work well in the offline setting, and in general, we are unaware of any prior work in offline MBO that provides offline tuning strategies. For fair comparisons to our results in Table 1, we report the performance of COMs after 50 training epochs that the authors in COMs found via online tuning, and find that IOM outperforms COMs.
> >
> > That said, we have now added one other strategy for offline tuning of COMs in **Figure 3 (right)**. This strategy, labeled as “two-step tuning” in Figure 3,  tunes COMs by first filtering out runs that attain a small enough value of the conservatism loss, and then picks the ones that attain the smallest validation prediction error. Observe in Figure 3 that this strategy does not still perform as well as our workflow for IOM, and attains a larger suboptimality compared to oracle tuning of COMs.
> >
> > We are happy to add other strategies for tuning COMs, if the reviewer might have any suggestions.
> >
> > ___
> >
> > > **computational complexity of tuning IOM**
> >
> > We have now added a discussion of the computational complexity of running IOM in **Appendix A.6**. Our tuning procedure should not lead to an extra overhead in time, since it only compares the loss values obtained from some training runs. Tuning every offline MBO method in our setup requires running the method for a fixed hyperparameter budget, and in general, we wouldn’t expect IOM to be significantly slower than other methods such as COMs for a given network size, since the only extra overhead in IOM comes from training an extra discriminator. We utilize only a 2-layer fully connected network for representing the discriminator which is pretty quick to train.
> >
> > ___
> >
> > **Please let us know if these answers and revisions address your concerns.**

---

> ### Author Response · Authors · 2022-08-05
> **Request for discussion**
>
> Dear Reviewer 1x47,
>
> We were wondering if you have gotten a chance to go through our responses and the revisions to the paper, and if these revisions and responses address your concerns regarding the paper. We are happy to address any remaining concerns and would really appreciate it if you engage in a discussion with us.
>
> Thank you so much!

---

> > ### Author Response · Authors · 2022-08-07
> > **Discussion**
> >
> > Dear Reviewer 1x47,
> >
> > There are only 2 more days remaining in the author-reviewer discussion period -- we were wondering if you have gotten a chance to go through our responses and the revisions to the paper. We are happy to address any remaining concerns and would really appreciate it if you engage in a discussion with us.
> >
> > Thank you so much!

---

> > > ### Comment · Reviewer_1x47 · 2022-08-07
> > > **Response to  response**
> > >
> > > Overall, i think the author's response  have addressed most of my concerns, and hence keep my score.

---

### Official Review · Reviewer_6jPy · 2022-07-11

**Rating:** 6
**Confidence:** 3
**Soundness:** 3 good
**Presentation:** 4 excellent
**Contribution:** 2 fair

**Summary:**

This paper studies the offline data-driven model-based optimization problem, where the dataset is static and doesn't allow for collecting new data. The main challenge is to prevent the optimizer from finding an out-of-distribution design. To achieve this, the authors cast this problem as a domain adaption problem and focus on representation learning. To deal with the out-of-distribution solutions, the authors learn an invariant representation which ensures the representation of out-of-distribution designs is uninformative for distinguishing good solutions from bad ones. As a result, the function value of out-of-distribution designs will stay close to the mean of the function values of training data. Consequently, it prevents the optimizer from selecting those designs. The authors also provided analysis, showing that the design found by their methods can compete favorably against the distributions of designs in the training dataset in the worst case. Empirically, the proposed method performs can attain improved designs over the best designs in the training data.

**Questions:**

See the strengths and weaknesses part.

**Limitations:**

See the strengths and weaknesses part.

**Strengths And Weaknesses:**

Strength:

1. The entire paper is well-structured and easy to follow. In particular, the method of learning an invariant representation to combat the out-of-distribution designs is very well-motivated. In constrat to methods that modify the constraints, the proposed method is compatible with nearly all the off-the-shelf optimizers.

2. The authors also provided many intuitive explanations for their method, which were very helpful for understanding the paper. For example, why will the invariance lead to "average" values for those out-of-distribution designs?

3. Though not significant, the authors provide a worst-case analysis for the performance of the method.

Weakness:

4. From my perspective, I feel that the method might be an overkill to the problem. Incorporating representation learning introduces extra but unnecessary complexity to the problem: as a result of the invariant representation, the function value of out-of-distribution designs will be penalized to be the "average" function values of the training data. So, why not directly learn an auxiliary function to penalize the function value of those out-of-distribution designs? This should be an easier task than learning an invariant representation.

5. The theoretical analysis is kind of straightforward from the existing literature. For example, similar results have already been shown in offline reinforcement learning (see Theorem 4.4), e.g., Yu et al., 2020.

Minor:

6. For Assumption 4.1, "there exists a representation \phi ..." seems redundant to min_{\phi, g}. I would  remove \min_{\phi, g}.

7. I didn't fully get "a novel connection with domain adaptation" (Line 51 - 52). Can you please highlight the "novel connection"?


Overall, I think this is an interesting work, and it might be interesting for people working on offline reinforcement learning. This paper provides some promise for attacking distribution-shift from the representation learning perspective.


Yu, Tianhe, Garrett Thomas, Lantao Yu, Stefano Ermon, James Y. Zou, Sergey Levine, Chelsea Finn, and Tengyu Ma. "Mopo: Model-based offline policy optimization." Advances in Neural Information Processing Systems 33 (2020): 14129-14142.

---

> ### Author Response · Authors · 2022-08-01
> **Author Response**
>
> We thank the reviewer for their detailed feedback and a positive assessment of this paper. We have updated the paper to address the minor comments, and answer the remaining questions below. The revisions to the paper are shown in $\textcolor{red}{red}$.
>
> ___
>
> > **Representation learning might be an overkill**
>
> This is a great question. The reviewer’s suggestion of the approach of penalizing the value of out-of-distribution designs is essentially similar to the conservative objective models (COMs) method from Trabucco et al. 2021, that we evaluate in our experiments. We find that IOM outperforms COMs in our experiments, and unlike COMs, the hyperparameters in IOM are amenable to be tuned via an offline workflow that we discuss in Section 5.
>
> To further understand the benefits of representation learning, as opposed to directly penalizing the learned value, we investigated the properties of the models learned by IOM and COMs in **Figures 8 & 9** in the Appendix.  We find that it is harder to minimize the training prediction error (i.e., error in accurately predicting the ground-truth value of a given design $\mathbf{x}$) with COMs as opposed to IOM. Moreover, COMs learn more non-smooth functions that IOM as the norm of the gradient with respect to the input, since the value of norm of the gradient of the learned model with respect to the input $\mathbb{E}_{\mathbf{x} \sim \mathcal{D}}[||\nabla_x f_\theta(\mathbf{x})||_2^2]$ is higher for COMs (Figure 9). This aligns with our intuition that an objective that attempts to directly modify the learned objective value may distort the learned objective function more than an objective on representations. Of course, a more complete analysis is needed to understand the relative tradeoffs of representation learning vs direct penalty in the output space and we have added a discussion of this direction in our future work, however, we believe that performance improvements and the efficacy of the offline workflow for IOM indicate that IOM is of interest to the community.
>
> ___
>
> > **theoretical analysis is kind of straightforward from the existing literature**
>
> We would like to clarify that the goal of our theoretical analysis is to simply show that invariant representation learning can be a useful tool for handling distributional shift in offline MBO, and not to develop new theoretical tools. We already cite and utilize techniques from some works in offline RL, and we have now revised the text to clearly indicate our goal with the theoretical analysis in Section 4.3. We have also cited the MOPO paper mentioned in the review.
>
> ___
>
> > **novel connection to domain adaptation**
>
> In domain adaptation, we are provided with a source and test domain, where the goal is to achieve good performance under the test domain, while training on the source domain. However, unlike this conventional use case, in our work we use domain adaptation to devise an invariance regularizer to handle distribution shift _within_ a single task / domain.
>
> As we discuss in the paper in Lines 144-152, offline MBO presents two distributions – the data distribution which we train on, and the distribution of the learned optimized designs, whose ground truth value determines the efficacy of a method. The novel connection is to view offline MBO as domain adaptation: our goal is to train on the data distribution, but attain good performance on the distribution of optimized designs. However, unlike standard domain adaptation, where the target distribution is fixed and given, in MBO, the optimization procedure controls the target distribution. Nevertheless, we show that despite this difference, techniques from domain adaptation can be effective in offline MBO, both in theory (Section 4) and in practice (Section 6).
>
> **Please let us know if these responses address your concerns. We are happy to clarify any further.**

---

> > ### Comment · Reviewer_6jPy · 2022-08-08
> > **Thanks**
> >
> > Thanks for the response. I don't have any additional concerns. I would like to keep the rating unchanged.

---

### Official Review · Reviewer_h8mh · 2022-07-11

**Rating:** 7
**Confidence:** 4
**Soundness:** 3 good
**Presentation:** 3 good
**Contribution:** 3 good

**Summary:**

This paper develops an offline model-based optimization method based on learning invariant representations, a popular approach in domain adaptation. The authors frame the invariant representation learning as a bi-level optimization problem and propose a practical algorithm for solving it. The authors also develop an effective offline hyperparameter selection procedure that's empirically shown to lead to the smallest sub-optimality gap. Experiment results show that the proposed offline MBO method outperforms previous approaches.

**Questions:**

1. Figure 1 is confusing. Why is conservatism shown to significantly overestimate the function value in the high-data region? Could the authors provide the reasoning on why this could happen?
2. In line 92, the authors state " IOM not only does not require a generative model" but in line 62-63 "we instantiate IOM using the $\chi^2$-discrepancy measure via a least-squares generative adversarial network". These two statements are conflicting with each other. Could the authors clarify?

**Limitations:**

The authors addressed the limitations.

**Strengths And Weaknesses:**

### Strength
1. The proposed offline model-based optimization method based on learning invariant representation is well justified and empirical results show strong performance.
2. The paper is well-written in general and most parts are easy to follow.
3. The paper discussed an offline tuning strategy that's critical for real-life applications.

### Weakness
1. The clarity of the paper could be further improved. Please see my comments in Questions section.

---

> ### Author Response · Authors · 2022-08-01
> **Author Response**
>
> We thank the reviewer for their detailed comments and for a positive assessment of this work. We have revised several parts of the paper to improve clarity and especially clarified how invariance enables us to tackle distributional shift via a newly-added visual illustration in **Figure 10 (Page 25)**. The revisions to the paper are shown in $\textcolor{red}{red}$. Additionally, we answer the questions in the review below:
>
> ___
>
> > **Why is conservatism shown to significantly overestimate the function value in the high-data region? Could the authors provide the reasoning on why this could happen?**
>
> Our model of conservatism is based on conservative objective models (COMs; Trabucco et al. 2021). This method attempts to not just push down the value of the learned objective model on out-of-distribution designs, but also attempts to push up the value of the in-distribution designs, observed in the training set. The illustration in Figure 1 captures this effect.
>
> However, we now clarify that this illustration is representative of the COMs method in particular, and other conservative approaches may not push up the value of points within the training distribution.
>
> ___
>
> > **IOM not only does not require a generative model**
>
> We have now updated the paper to clarify this. IOM does not require a generative model over the design space of $\mathbf{x}$ during optimization: unlike prior methods like CbAS (Brookes et al. 2019) and MINs (Kumar et al. 2020) that use VAE and GANs over the design space, IOM can simply find optimized designs using gradient descent.
>
> As the reviewer notes, the invariance regularizer is implemented using the discriminator that attempts to discriminate representations for training and OOD designs. While this design is inspired from a generative adversarial network, which is a generative model, IOM does not utilize any generator that directly attempts to produce a new design in the space of $\mathbf{x}$, which may be extremely large. The discriminator only attempts to discriminate **representations**, which are substantially lower-dimensional than the input space (5000 original dimension for HopperController vs <= 1024 dimensional representations).
>
> Therefore, IOM does not require a generative model over the design space. We have updated this in the paper to clearly explain this.

---

### Official Review · Reviewer_Jwpc · 2022-07-13

**Rating:** 6
**Confidence:** 3
**Soundness:** 3 good
**Presentation:** 3 good
**Contribution:** 3 good

**Summary:**

 This work studies offline data-driven model based optimization (MBO), where the goal is to produce optimized designs, purely using a static dataset. We proposed to view offline data driven model-based optimization through the lens of domain adaptation, and proposed a method that trains a model of the objective with an additional regularizer to promote invariance between the representations learned by the model in expectation under the training data distribution and the distribution of optimized points.

**Questions:**

Please refer to "Weakness".

**Ethics Review Area:**

["I don’t know"]

**Limitations:**

They have adequately addressed the limitations and potential negative societal impact of their work.

**Strengths And Weaknesses:**

Strengths:

1.This paper studies the policy learning problem under the distributional shift. The total goal is to learn an invariant policy (representation) which maximizes some specific objectives (specific rewards) in the varying environments. Some related work is provided to support their novelty.

2.Since the authors assumes the variability of marginal distributions (e.g., covariate shifts), they proposed the IOM method with MMD divergence between training and target features. Some theoretical results are provided to show the performance guarantee of the IOM method (a lower bound).

3.Extensive experiments are provided to verify the effectiveness of the proposed method. Some ablation studies are provided with details.

Weaknesses:

1.I feel confused on the novelty of the proposed method. From my viewpoint, this paper presents a policy learning method with distributional shift. However, offline policy learning methods with distributional shift is already a popular topic. Moreover, it is not rare to use some DA tricks to improve the transferability of the learned policy across different environments.

2.Related work should distinguish this paper from previous policy learning methods. Their motivations are settings are very similar.

---

> ### Author Response · Authors · 2022-08-01
> **Author Response**
>
> We thank the reviewer for their detailed feedback and a positive assessment of this work. We have attempted to clarify the novel contributions of this work, and clarified the distinction against prior works. **Please let us know if this addresses the concerns. We would be happy to answer any more questions.**
>
> >  **I feel confused on the novelty of the proposed method. From my viewpoint, this paper presents a policy learning method with distributional shift. However, offline policy learning methods with distributional shift is already a popular topic; motivations and settings are very similar.**
>
> While many prior works look at policy learning and transfer learning (where the policy must handle distributional shift at test time, arising for example from being tested in a new domain), our use of domain adaptation is different in a subtle but important way: we are not proposing a better transfer learning method to handle distributional shift due to a change in domain, but rather a way to mitigate the out-of-distribution inputs problem, which afflicts all offline optimization methods (including offline RL and offline MBO) by means of domain adaptation methods. To our knowledge, no prior work in either offline RL or offline MBO takes this domain adaptation perspective to devise an invariance regularizer on the learned objective / value function.
>
> Specifically, our experimental results show that IOM outperforms prior offline MBO methods that attempt to solve the very same distributional shift problem. In addition, we also propose an effective offline strategy for tuning IOM, unlike prior methods (such as COMs) which do not specify a workflow and are harder to tune. We believe that the problem of policy learning with distributional shift is not entirely solved and better approaches for offline MBO are still of interest to the community.
>
> We would be happy to add comparisons to and discuss any prior methods that the reviewer considers particularly relevant in this regard, or add discussion of these works to the paper, but we believe that we covered the closest related works that are feasible to compare to our approach.
>
> ___
>
>
> > **Moreover, it is not rare to use some DA tricks to improve the transferability of the learned policy across different environments**.
>
> We think perhaps there might be a misunderstanding and would like to clarify. We do not use domain adaptation (DA) tricks to improve transferability of the learned policy across different environments, instead we use domain adaptation in a subtle way: the DA perspective is used to tackle the challenge of distributional shift **within** the same task / environment. This is not the usual way in which DA is used. The DA perspective gives us an invariance regularizer that enables us to control the learned policy from deviating too far away from the training policy. We believe that this kind of use of DA in tackling distributional shift within a single environment / task does not appear in prior works in offline MBO, and is hence novel and of interest to the community.
>
> That said, we are happy to discuss any prior works in DA we missed that the reviewer considers particularly relevant in this regard, or add discussion of these works to the paper. Are there any particular papers that you have in mind that use domain adaptation in this way?

---

> > ### Comment · Reviewer_Jwpc · 2022-08-08
> > **Thanks for your responses**
> >
> > I am satisfied with your responses and my concerns are mostly addressed. I am willing to keep my score.

---

### Author Response · Authors · 2022-08-02
**Summary of Changes (as of August 2)**

Dear Reviewers and the AC,

Thank you for the detailed feedback on our paper. We have responded to your individual comments, and made revisions in the paper. The changes to the paper are shown in $\textcolor{red}{red}$. In this post, we will summarize the major changes

1. Added an additional baseline that maximizes the lower-confidence bound obtained from a GP posterior in Table 2 (LCB + GP) [Reviewer VgHW]

2. Added Figure 10, Appendix A.7 to describe a schematic of how invariance works [Reviewer VgHW]

3. Added results comparing training error and smoothness of COMs and IOM to understand why invariance can perform better in Appendix A.6 [Reviewers 6jPy, 1x47]

4. Added clarification of baselines (baselines are taken directly from the Design bench benchmark (Trabucco et al. 2022), and clarified that we already compare to the best point in the dataset [Reviewer VgHW]

5. Discussed computational cost of tuning IOM in Appendix A.6 [Reviewer 1x47]

6. Evaluated a two-stage tuning strategy for COMs in Figure 3 [Reviewer 1x47]

7. Added extension of theoretical result to handle comparisons against the best point in the dataset [Reviewer VgHW]

8. Rewritten the paragraph the intuition behind IOM [Reviewer VgHW] and the problem of distributional shift

9. Clarified relation to related works in domain adaptation and policy learning (and novelty of this work) [Reviewer Jwpc] and other robust BO works [Reviewer VgHW]

10. Clarified the questions regarding Figure 1, and IOM not requiring a generative model [Reviewer h8mh]

Additionally, we have also attempted to make other small changes, and answered the reviewers' questions directly as responses to their review. Please let us know if these address your concerns. We are happy to clarify any remaining concern.

Thank you so much, and looking forward to the discussion!

---

### Meta-Review · Area_Chair_KaYZ · 2022-08-26

**Recommendation:** Accept
**Confidence:** Certain

**Metareview:**

Reviewers are all positively inclined for the paper. Most of them have stated that the paper
proposes some novel and relevant ideas with theoretical support and empirical evidences
of theirs soundness. As such, we think that the paper can be accepted to the conference, and
expect some revisions that would improve clarity of the work.


**Award:**

No

---

### Decision · Program_Chairs · 2022-09-14

Accept